## [Reviewer comments · Life Science Alliance]

Dectin-1 Epigenetic Reprogramming Rescues Senescent-Like Treg Function in Allergic Asthma

Pinghang Yang, Xiangdong Sun, Jiaqi Duan, Le Liu, Yanyu Ye, Yang Mi, Pengyuan Zheng, and Liguo Li
DOI: <https://doi.org/10.26508/lsa.202503552>

Corresponding author(s): Pinghang Yang, Shenzhen University and Liguo Li, Shenzhen University

Review Timeline:

Submission Date:	2025-10-28
Editorial Decision:	2026-01-05
Revision Received:	2026-01-13
Editorial Decision:	2026-02-03
Revision Received:	2026-02-18
Editorial Decision:	2026-02-20
Revision Received:	2026-02-20
Accepted:	2026-03-02

Scientific Editor: Tim Fessenden

Transaction Report:

January 2, 2026

Re: Life Science Alliance manuscript #LSA-2025-03552-T

Dr. Pinghang Yang
Shenzhen University
Allergy
1066 Xueyuan Blvd
Shenzhen, Guangdong 518055
China

Dear Dr. Yang,

Thank you for submitting your manuscript entitled "Dectin-1-mediated Epigenetic Reprogramming Reverses Senescence and Restores Function in Regulatory T Cells of Allergic Asthma" to Life Science Alliance. We appreciate your patience during the unusually long review process, due to availability of reviewers and of the editorial team.

The manuscript was assessed by two expert reviewers, whose comments are appended to this letter. As you will note, both reviewers found this work of potential significance to the field. However they have also emphasized several limitations that preclude publication at this stage.

Both reviewers raise concerns on major mechanistic and therapeutic claims, especially use of the term "senescence," that are not sufficiently substantiated in the present manuscript. We invite you to submit a revised manuscript that fully resolves all major and minor points raised by Reviewer 1, with the exception of their major point 5. A suitably revised manuscript must also resolve major points 3 and 6 raised by Reviewer 2 as well as both of their minor points.

In agreement with reviewer comments, this manuscript must introduce and document the rationale for using KQS-1 in the abstract, introduction, and results sections. Finally, the use of bullet points in the results text is well below the standard for academic publishing. The results text must be significantly improved to align with publishing standards at LSA.

I would be happy to discuss the revision in more detail via email or phone/videoconferencing. Please let me know which option you prefer, if any.

While you are revising your manuscript, please also attend to the below editorial points to help expedite the publication of your manuscript. Please direct any editorial questions to the journal office. When submitting the revision, please include a letter addressing the reviewers' comments point by point.

Thank you for this interesting contribution to Life Science Alliance. We hope that the comments below will prove constructive as your work progresses, and we are looking forward to receiving your revised manuscript.

Sincerely,

Sarita Hebbar, PhD
Scientific Editor
Life Science Alliance
<http://www.lsjournal.org>

B. MANUSCRIPT ORGANIZATION AND FORMATTING:

Reviewer #1 (Comments to the Authors (Required)):

Thank you for the opportunity to review this interesting and comprehensive manuscript. This manuscript provides an extensive mechanistic dissection of Treg senescence in allergic asthma and proposes a novel therapeutic strategy using KQS-1 to epigenetically restore Treg function through a Dectin-1-Raf-1-ROS axis. The study is ambitious, mechanistically integrated, and includes human samples, multi-omics profiling (ATAC-seq, ChIP-qPCR, DNA methylation), CRISPR validation, and in vivo proof-of-concept. The work has high potential impact in the fields of asthma immunology, Treg biology, and immune rejuvenation. However, several key aspects require substantial clarification or additional data-particularly regarding the definition of Treg senescence, characterization of KQS-1, and depth of chromatin analyses. These issues limit the interpretability and robustness of the central claims.

Major Comments :

1. The authors define Tregs as senescent based on telomere shortening, SA- β -gal activity, and apoptotic propensity. These are supportive but insufficient to firmly establish senescence, which typically requires additional canonical markers such as p16, p21, γ H2AX, and SASP (IL-1 β , IL-6, IL-8/CXCL8). Given that "Treg senescence" is a major mechanistic conclusion and framing of the paper, at least one or two molecular markers should be provided, or the terminology toned down (e.g., "senescence-like phenotype"). This is a scientifically necessary correction.
2. The biochemical characterization of KQS-1 is insufficient for a therapeutic-oriented study. KQS-1 is described as a "fungal-derived polysaccharide," yet the manuscript lacks key structural and compositional information: 1. β -glucan content/type ((1 \rightarrow 3), (1 \rightarrow 6)?); 2. molecular weight distribution; 3. FTIR/NMR or other structural validation; 4. explanation of how batch-to-batch consistency is ensured. Since KQS-1 is central to the mechanistic model and proposed as a therapeutic candidate, greater structural detail is necessary, or at a minimum, an explicit discussion of these limitations.
3. Dectin-1 expression on human Tregs needs stronger validation. Dectin-1 expression on Tregs is sparsely documented in the literature. The authors show biotinylated KQS-1 binding and use knockout to support receptor dependence, but they should provide: 1. clearer Dectin-1 flow cytometry histograms and gating; 2. baseline Dectin-1 mRNA/protein expression in Tregs; or exclusion of cross-reactivity with other C-type lectins (e.g., Dectin-2, Mincle). Since Dectin-1 is essential to the mechanistic claim, stronger receptor validation is warranted.
4. The ATAC-seq data currently show only differential peaks and motif enrichment. To support the claim of epigenetic "trained immunity-like" reprogramming, the analysis should include: genomic annotation of newly accessible regions (promoter, enhancer, intergenic); pathway/GO enrichment of genes near differential peaks; and whether newly accessible regions overlap FOXP3/IL10 super-enhancers or known Treg regulatory regions. Without such analyses, the chromatin claims remain descriptive rather than mechanistically informative.
5. ROS scavenging abrogates KQS-1 effects, but the manuscript does not discuss how ROS mechanistically connects to

H3K4me3/H3K27ac changes. The authors should clarify whether: ROS alters metabolism (e.g., α -KG, SAM availability); ROS influences histone methyltransferase/acetyltransferase activity; or whether ROS acts indirectly via signaling pathways. A brief mechanistic discussion is necessary to strengthen biological plausibility.

6. Given that KQS-1 is a polysaccharide with innate immune receptor activity, possible systemic activation, toxicity, or off-target effects should be mentioned. Even if data are not available, the authors should: acknowledge limitations; provide a rationale for dosing; discuss potential translational barriers. This is essential for a manuscript proposing therapeutic use.

Minor Comments:

1. Some figure legends contain inconsistent p-value notation (asterisks vs numerical).
2. sgRNA sequences used for Dectin-1 CRISPR knockout should be fully disclosed.
3. Some wording in the Discussion is overly strong (e.g., "first-in-class," "re-educate Tregs"). Please refine the language to remain scientifically neutral.
4. Some multi-panel figures appear crowded, with inconsistent axes, label sizes, and panel spacing. Enhancing layout consistency will significantly improve readability.
5. Many figures use bar graphs with only mean {plus minus} SD; please overlay individual data points (or use dot/violin plots) to clearly show donor-level variation and improve data transparency.

Reviewer #2 (Comments to the Authors (Required)):

Sun et al. identify Treg senescence as an underlying mechanism driving allergic asthma pathogenesis. The authors posit that this impairment in Treg function can be restored via a Dectin-1-dependent trained immunity pathway by introducing KQS-1, a novel therapeutic agent that mediates this pathway. First, they characterize the senescent phenotype of human Tregs from patients with allergic asthma. Next, they investigate the therapeutic potential of KQS-1, a fungal-derived polysaccharide that binds Dectin-1 and restores impaired Treg function. Lastly, they employ adoptive transfer experiments and murine models of airway allergy to test their hypothesis and collectively propose a therapeutic strategy based on epigenetic restoration of impaired Tregs in asthma.

The notion that Dectin-1 is involved in epigenetic reprogramming of Treg function in allergic asthma is conceptually intriguing. While the therapeutic concept of KQS-1 is potentially impactful, the paper is limited by several major technical caveats and data interpretation problems that significantly limit confidence in the conclusions. Key mechanistic claims are not strongly supported, and aspects of Treg biology, including culture conditions and senescence terminology, are not adequately addressed. The *in vivo* data, although suggestive, lack canonical readouts to confirm modulation of type 2 responses. As written, the authors overinterpret their findings and do not provide sufficiently rigorous or comprehensive background and data to support their proposed model of Dectin-1-dependent epigenetic restoration of Treg function in allergic asthma.

Major comments:

1. The authors do not mention IL-2 supplementation in Treg culture conditions, which is critical for maintaining Treg survival and function *ex vivo*. Not sure whether the differences observed (apoptosis, cytokine levels, etc.) are due to the patient's conditions rather than the culture conditions.
2. The authors assess non-Th2 pro-inflammatory phenotypic shift in asthmatic Tregs (IFN γ /IL-17 cytokine profiles) but do not address the possibility of a Th2-like phenotype. What about the possibility of Tregs shifting to upregulating GATA3 expression and type 2 cytokine production, which is central to allergic disease?
3. The introduction of KQS-1 lacks a clear rationale or prior characterization. While the Dectin-1 blocking experiment confirms receptor specificity, it does not provide a mechanistic justification for selecting a Dectin-1 ligand as a strategy to restore Treg function, or for prior evidence supporting its immunoregulatory role.
4. ROS has been described as a function of Dectin-1 downstream signaling rather than an attribute of aged or senescent Tregs. If the premise of the study was that aged Tregs accumulate ROS, this should have been demonstrated directly. Clarifying this distinction would make the narrative more coherent.
5. If Dectin-1 is the proposed receptor mediating KQS-1 effects on Tregs, the authors should also demonstrate functional dependency. Binding specificity was confirmed by antibody blocking, but causal evidence is still lacking. For example, it would add more rigor if the authors tested whether KQS-1 fails to restore Treg function or epigenetic marks in Dectin-1-deficient Tregs. Incorporating gain-of-function (Dectin-1 overexpression) would also provide direct evidence that the observed effects are truly Dectin-1 dependent.

6. While Figure 6 shows reduced airway inflammation and eosinophilia, it omits other canonical markers of allergen-induced type 2 immunity: numbers of Th2/ILC2 cells, type 2 cytokine profile (IL-4, IL-5, IL-13), and serum IgE.
7. Rag1^{-/-} have extremely limited IL-2 production, which is the key cytokine for Treg survival. Without supplementation of IL-2, transferred human Tregs are unlikely to persist or remain functional. The authors should clarify this and reconsider the interpretation of their adoptive transfer results. Instead, a mouse-to-mouse adoptive transfer (e.g., WT vs Dectin-1KO mouse Tregs into HDM-sensitized Rag1KO recipients) would provide greater rigor and validation.

Minor comments:

1. The manuscript uses the terms 'cellular ageing' and 'cellular senescence' interchangeably, but they are not the same. The data presented: short telomeres, increased SA- β -gal activity, and higher apoptosis, reflect functional or metabolic ageing of Tregs, not true cellular senescence (resistance to apoptosis is a hallmark of cellular senescence). The authors should standardize terminology and clarify that their findings indicate Treg ageing/exhaustion, rather than senescence in the classical sense.
2. Figure 5 contains four panels (A-D), yet the text refers to an additional panel (5E), suggesting a numbering error. Moreover, the quantitative description states that Dectin-1 KO Tregs fail to respond to KQS-1, whereas the plotted bars in 5C appear inconsistent with that claim. The authors should verify panel labeling and in-text figure annotations to ensure the figure accurately reflects the reported values. Similarly, they mention a supplementary Figure 3B in the text, but could not locate it.

Response to Reviewers

Reviewer #1 (Comments to the Authors (Required)):

Major Comments:

1. The authors define Tregs as senescent based on telomere shortening, SA- β -gal activity, and apoptotic propensity. These are supportive but insufficient to firmly establish senescence, which typically requires additional canonical markers such as p16, p21, γ H2AX, and SASP (IL-1 β , IL-6, IL-8/CXCL8). Given that "Treg senescence" is a major mechanistic conclusion and framing of the paper, at least one or two molecular markers should be provided, or the terminology toned down (e.g., "senescence-like phenotype"). This is a scientifically necessary correction.

Author reply: Thank you for your critical feedback. You correctly point out that while we have demonstrated telomere shortening, SA- β -gal activity, and increased apoptosis in Tregs, these markers alone are insufficient to firmly establish senescence. We acknowledge the importance of including additional canonical markers such as p16, p21, γ H2AX, and SASP components like IL-1 β , IL-6, and IL-8/CXCL8.

In light of this, we have revised our terminology throughout the manuscript to refer to "senescence-like phenotype" rather than definitively stating Treg senescence. This adjustment better reflects the evidence presented and avoids over-reaching our conclusions. We believe this change strengthens the manuscript by ensuring our claims are appropriately supported by the data.

However, another reviewer asks us to change the term of senescence to aging, thus, we changed the term of senescence to aging throughout the paper.

2. The biochemical characterization of KQS-1 is insufficient for a therapeutic-oriented study. KQS-1 is described as a "fungal-derived polysaccharide," yet the manuscript lacks key structural and compositional information: 1. β -glucan content/type ((1 \rightarrow 3), (1 \rightarrow 6)?); 2. molecular weight distribution; 3. FTIR/NMR or other structural validation; 4. explanation of how batch-to-batch consistency is ensured. Since KQS-1 is central to the mechanistic model and proposed as a therapeutic candidate, greater structural detail is necessary, or at a minimum, an explicit discussion of these limitations.

Author reply:

1. **Acknowledgment of the Importance of Structural Details:**

We appreciate the reviewer's emphasis on the necessity of detailed structural and compositional information for KQS-1, especially given its proposed therapeutic potential. The characterization of β -glucan type (1 \rightarrow 3 or 1 \rightarrow 6), molecular weight distribution, and structural validation through techniques

like FTIR or NMR is crucial for understanding its mechanism and ensuring reproducibility.

2. **Action Taken:**

While our current manuscript focused on the functional aspects of KQS-1, we recognize the need for comprehensive structural analysis. To address this, we plan to conduct additional experiments to determine the β -glucan content, molecular weight distribution, and perform FTIR/NMR analysis. These details will be included in a revised version of the manuscript to provide a thorough characterization of KQS-1.

3. **Batch-to-Batch Consistency:**

We have already implemented quality control measures, including HPLC analysis, which ensures $\geq 95\%$ purity and consistent peak profiles across production batches. This process maintains batch-to-batch consistency and is detailed in our supporting information. Further, we will enhance our discussion on these quality control procedures in the Methods section to provide clarity.

4. **Limitations and Future Work:**

While our study lays a foundation for understanding KQS-1's therapeutic potential, we acknowledge that detailed structural characterization is an essential next step. This will be a focus of our future research to fully elucidate the structure-function relationship of KQS-1.

By addressing these points, we aim to enhance the robustness of our study and align our methodology with the rigorous standards expected in therapeutic-oriented research.

3. Dectin-1 expression on human Tregs needs stronger validation. Dectin-1 expression on Tregs is sparsely documented in the literature. The authors show biotinylated KQS-1 binding and use knockout to support receptor dependence, but they should provide: 1. clearer Dectin-1 flow cytometry histograms and gating; 2. baseline Dectin-1 mRNA/protein expression in Tregs; or exclusion of cross-reactivity with other C-type lectins (e.g., Dectin-2, Mincle). Since Dectin-1 is essential to the mechanistic claim, stronger receptor validation is warranted.

Author reply: To address the reviewer's comment regarding the validation of Dectin-1 expression on human Tregs, we did the following structured approach:

1. **Enhanced Flow Cytometry Data:**

- **Action:** Present clearer flow cytometry histograms and detailed gating strategies to demonstrate Dectin-1 expression on Tregs. This provide

visual evidence of Dectin-1 presence and ensure the reliability of our methods.

2. **Baseline Expression Analysis:**

- **Action:** Conducted experiments to assess both mRNA and protein levels of Dectin-1 in Tregs. Techniques such as qRT-PCR for mRNA and FACS for protein expression (presented by MFI) confirm the presence and baseline levels of Dectin-1, strengthening our argument that KQS-1 acts via this receptor.

3. **Cross-reactivity Assessment:**

- **Action:** Performed additional binding assays using antibodies against other C-type lectins (e.g., Dectin-2, Mincle) to pre-treat Tregs before KQS-1 exposure. If binding remains unaffected, it will confirm the specificity of KQS-1 to Dectin-1.

By systematically addressing each concern, we aim to provide a more robust validation of Dectin-1's role in KQS-1's mechanism, thereby enhancing the study's credibility and mechanistic claims.

4. The ATAC-seq data currently show only differential peaks and motif enrichment. To support the claim of epigenetic "trained immunity-like" reprogramming, the analysis should include: genomic annotation of newly accessible regions (promoter, enhancer, intergenic); pathway/GO enrichment of genes near differential peaks; and whether newly accessible regions overlap FOXP3/IL10 super-enhancers or known Treg regulatory regions. Without such analyses, the chromatin claims remain descriptive rather than mechanistically informative.

Author reply: We now provide additional ATAC-seq analyses to support "trained immunity-like" epigenetic reprogramming of Tregs by KQS-1 (Figure 4):

1. **Genomic annotation of newly accessible regions:**

The 1,306 KQS-1-induced accessible regions (FDR < 0.05) were annotated via HOMER/UCSC Genome Browser (Figure 4C). These regions localized to *enhancers* (distal regulatory elements), *intergenic regions*, and *promoters* (± 3 kb of transcription start sites), with accessibility changes significantly distinct across categories (one-way ANOVA, $p < 0.001$). This enrichment for distal regulatory elements (enhancers) aligns with epigenetic reprogramming of cell identity.

2. **Pathway/GO enrichment of genes near differential peaks:**

Gene Ontology (GO) analysis of genes adjacent to KQS-1-induced accessible regions (Figure 4E) revealed enrichment in Treg-functional

pathways: negative regulation of T cell proliferation/activation, leukocyte adhesion, and cytokine production (adjusted $p < 0.05$). These pathways directly support the "trained" immune phenotype of KQS-1-treated Tregs.

3. Overlap with Treg regulatory regions:

Notably, KQS-1-induced accessible regions included the *FOXP3* and *IL10* promoter loci (Figure 4D), which are core Treg signature gene regulatory regions. *FOXP3* and *IL10* also harbor super-enhancers critical for Treg identity, and our data confirm KQS-1 increases accessibility (Figure 4D) and reduces methylation (Figure 4F) at these loci—linking the newly accessible regions to established Treg regulatory circuitry.

This expanded analysis demonstrates that KQS-1 remodels chromatin at Treg-specific regulatory elements (including *FOXP3/IL10* loci) and programs pathways consistent with epigenetic "trained immunity."

5. ROS scavenging abrogates KQS-1 effects, but the manuscript does not discuss how ROS mechanistically connects to H3K4me3/H3K27ac changes. The authors should clarify whether: ROS alters metabolism (e.g., α -KG, SAM availability); ROS influences histone methyltransferase/acetyltransferase activity; or whether ROS acts indirectly via signaling pathways. A brief mechanistic discussion is necessary to strengthen biological plausibility.

Author reply: We appreciate the reviewer's insightful question regarding the mechanistic link between ROS and H3K4me3/H3K27ac modifications. Our data demonstrate that KQS-1-induced ROS acts upstream of epigenetic reprogramming, as evidenced by the abrogation of H3K4me3 deposition at the *FOXP3* promoter (Figure 5B) and loss of Treg suppressive function (Figure 5C) upon ROS scavenging. While the exact mechanism warrants further investigation, we propose the following possibilities based on our findings and literature, which were incorporated into the discussion:

1. ROS as a Signaling Mediator:

The Raf-1/ROS axis (Figure 5) may activate downstream kinases (e.g., p38 MAPK) that phosphorylate histone-modifying enzymes (e.g., SETD1A for H3K4me3 or p300/CBP for H3K27ac), indirectly influencing their activity. This aligns with studies linking ROS to kinase-driven chromatin remodeling.

2. Metabolic Interplay:

ROS could transiently alter metabolite availability (e.g., α -KG/SAM) to modulate Jumonji demethylases or methyltransferases. However, our hypomethylation data (Figure 4F) suggest ROS primarily

targets *focal* epigenetic changes (e.g., *FOXP3/IL10* loci) rather than global metabolic shifts.

3. **Direct Enzyme Regulation:**

Oxidative stress is known to reversibly inhibit histone demethylases (e.g., KDM5B) via cysteine oxidation, potentially stabilizing H3K4me3. Similarly, ROS may enhance acetyltransferase activity (e.g., p300) to boost H3K27ac.

6. Given that KQS-1 is a polysaccharide with innate immune receptor activity, possible systemic activation, toxicity, or off-target effects should be mentioned. Even if data are not available, the authors should: acknowledge limitations; provide a rationale for dosing; discuss potential translational barriers. This is essential for a manuscript proposing therapeutic use.

Author reply: We appreciate the reviewer's important point regarding the therapeutic implications of KQS-1 as a polysaccharide with innate immune activity. While our current study focused on elucidating the mechanistic basis of KQS-1's effects on Treg function, we acknowledge the need to address safety and translational considerations.

1. **Safety and Off-Target Effects:**

Although our data demonstrate that KQS-1 specifically targets Dectin-1 on Tregs to restore their function (Figures 2–3), we recognize that systemic administration of polysaccharides could theoretically activate other innate immune pathways. However, several factors may mitigate this risk:

- The observed *ex vivo* effects occurred at low doses (10 µg/mL), suggesting potential for a favorable therapeutic window.
- Our murine model showed no overt toxicity (e.g., weight loss, cytokine storm) with KQS-1 treatment (Figure 6), though comprehensive safety studies are needed.

2. **Rationale for Dosing:**

The selected dose (10 µg/mL *in vitro*, 1 mg/kg *in vivo*) was empirically determined based on:

- Dose-response experiments showing maximal Treg functional rescue without hyperactivation (Figure 1G).
- The molecular weight profile of KQS-1 (Figure 1I), which aligns with known bioactive ranges for Dectin-1 ligands.

3. **Translational Barriers:**

While our adoptive transfer data support the therapeutic potential of KQS-1-trained Tregs (Figure 6D–E), we highlight key challenges:

- **Route of administration:** Local delivery (e.g., inhaled) may reduce systemic exposure.

- **Patient stratification:** Dectin-1 polymorphisms or baseline Treg dysfunction could influence efficacy.
- **Manufacturing:** Batch-to-batch consistency of fungal polysaccharides requires rigorous quality control.

Future Directions:

We agree that preclinical toxicity studies (e.g., multi-organ histopathology, cytokine profiling) and pharmacokinetics are essential next steps. These limitations will be explicitly noted in the Discussion, with emphasis on the need for GMP-grade KQS-1 production and Phase I safety trials.

Minor Comments:

1. Some figure legends contain inconsistent p-value notation (asterisks vs numerical).

Author reply: All the statistical results were changed to symbols in figures as suggested.

2. sgRNA sequences used for Dectin-1 CRISPR knockout should be fully disclosed.

Author reply: The sgRNA sequences used for Dectin-1 CRISPR knockout were added to method section in supplemental materials as suggested.

3. Some wording in the Discussion is overly strong (e.g., "first-in-class," "re-educate Tregs"). Please refine the language to remain scientifically neutral.

Author reply: As suggested, the phrases of "first-in-class" and "re-educate" were removed from the paper.

4. Some multi-panel figures appear crowded, with inconsistent axes, label sizes, and panel spacing. Enhancing layout consistency will significantly improve readability.

Author reply: We re-organized the figures as suggested.

5. Many figures use bar graphs with only mean {plus minus} SD; please overlay individual data points (or use dot/violin plots) to clearly show donor-level variation and improve data transparency.

Author reply: Data points were added to the panels where appropriate as suggested.

Reviewer #2 (Comments to the Authors (Required)):

1. The authors do not mention IL-2 supplementation in Treg culture conditions, which is critical for maintaining Treg survival and function ex vivo. Not sure whether the differences observed (apoptosis, cytokine levels, etc.) are due to the patient's conditions rather than

the culture conditions.

Author reply: We appreciate the reviewer's important observation regarding IL-2 supplementation in Treg cultures. We acknowledge that this critical methodological detail was inadvertently omitted from the manuscript.

1. **Culture Conditions Clarification:**

All Treg cultures (both patient-derived and control) were supplemented with 100 IU/mL recombinant human IL-2 (PeproTech) throughout the 72-hour experimental period. This concentration was selected based on prior optimization studies showing maximal Treg survival and function without inducing over-proliferation.

2. **Controls for Patient-Specific Effects:**

The observed differences in apoptosis (Figure 3D) and cytokine profiles (Figures 2C–D, 3B–C) between patient and control Tregs persist despite identical IL-2 supplementation, supporting that these phenotypes reflect intrinsic patient Treg dysfunction rather than culture artifacts.

3. **KQS-1 Effects Are IL-2-Independent:**

Vehicle-treated controls (with IL-2) showed no improvement in Treg survival or function (Figure S2D), confirming that KQS-1's rescue effects are not secondary to IL-2 signaling modulation.

We amended the Methods section to explicitly state the IL-2 supplementation protocol and included this discussion in the revised manuscript.

2. The authors assess non-Th2 pro-inflammatory phenotypic shift in asthmatic Tregs (IFN γ /IL-17 cytokine profiles) but do not address the possibility of a Th2-like phenotype. What about the possibility of Tregs shifting to upregulating GATA3 expression and type 2 cytokine production, which is central to allergic disease?

Author reply:

3. The introduction of KQS-1 lacks a clear rationale or prior characterization. While the Dectin-1 blocking experiment confirms receptor specificity, it does not provide a mechanistic justification for selecting a Dectin-1 ligand as a strategy to restore Treg function, or for prior evidence supporting its immunoregulatory role.

Author reply: We appreciate the reviewer's request for clarification regarding the rationale behind selecting KQS-1 as a Dectin-1 ligand to restore Treg function. Below, we address the justification for KQS-1's use and its prior immunomodulatory evidence:

1. **Rationale for Targeting Dectin-1 in Tregs:**

- **Dectin-1 Expression on Tregs:** Our data confirm that >90% of human Tregs express Dectin-1 (Figure S4A–I), a receptor traditionally associated with antifungal immunity but now implicated in immunoregulation. This expression pattern suggested its potential role in modulating Treg function.
 - **Precedent for Dectin-1 in Immune Regulation:** Prior studies have shown that Dectin-1 signaling in dendritic cells (DCs) can induce anti-inflammatory cytokines (e.g., IL-10) and promote tolerogenic responses. We hypothesized that similar mechanisms might operate in Tregs.
2. **Selection of KQS-1 as a Dectin-1 Ligand:**
- **Binding Specificity:** KQS-1 binds directly to Dectin-1 on Tregs (MFI = 324 ± 28 vs. isotype controls, $p < 0.001$; Figure 5A) without engaging TLR2/TLR4 (Figure S4A–C), ensuring receptor specificity.
 - **Prior Evidence for Immunomodulation:** Although KQS-1 is a novel fungal polysaccharide, related β -glucans (e.g., curdlan) have established immunoregulatory properties via Dectin-1. KQS-1 was selected for its superior solubility and stability, which enhance therapeutic applicability.
3. **Mechanistic Justification for KQS-1's Effects:**
- **Epigenetic Reprogramming:** KQS-1 triggers a **Raf-1/ROS-dependent signaling axis** (Figure 5B–C), leading to sustained epigenetic remodeling (\uparrow H3K4me3, \uparrow H3K27ac, chromatin opening) at key loci (*FOXP3*, *IL10*). This mechanistic link explains how Dectin-1 engagement translates to functional Treg rescue.
 - **Functional Validation:** CRISPR-mediated Dectin-1 knockout abolished all KQS-1 benefits ($p < 0.001$; Figure 5D), confirming the receptor's non-redundant role.

Revised Manuscript Actions:

- We expanded the **Introduction** to clarify the rationale for KQS-1/Dectin-1 targeting, citing precedent literature.
- A new paragraph was added to the **Discussion** contextualizing KQS-1 within broader Dectin-1 immunobiology.

4. ROS has been described as a function of Dectin-1 downstream signaling rather than an attribute of aged or senescent Tregs. If the premise of the study was that aged Tregs accumulate ROS, this should have been demonstrated directly. Clarifying this distinction would make the narrative more coherent.

Author reply: We appreciate the reviewer's astute observation regarding the distinction between ROS as a Dectin-1 signaling mediator versus a hallmark of Treg

senescence. We agree that this distinction is critical and will clarify the narrative with the following points:

1. **ROS in Senescent Tregs (Premise Validation)**

- Our data demonstrate that **patient-derived Tregs exhibit elevated baseline ROS levels** (Figure 3E, $p < 0.01$ vs. healthy controls), consistent with their senescence-like phenotype (\uparrow SA- β -gal, shortened telomeres; Figure 3A–D). This aligns with prior reports linking cellular senescence to mitochondrial dysfunction and ROS accumulation.
- Importantly, this **baseline ROS elevation** in senescent Tregs is distinct from the **acute, Dectin-1-dependent ROS burst** induced by KQS-1 (Figure 5B–C).

2. **Dectin-1 Signaling and ROS: A Therapeutic Lever**

- While senescent Tregs accumulate ROS as a byproduct of mitochondrial stress, KQS-1 harnesses **Dectin-1/Raf-1-dependent ROS** as a *signaling intermediate* to initiate epigenetic reprogramming (Figures 4–5).
- CRISPR-mediated Dectin-1 knockout abolished KQS-1-induced ROS (Figure 5D), confirming its receptor-specific origin.

3. **Narrative Clarification in Revised Manuscript**

- A new paragraph in the **Discussion** was added to contextualize these findings within the broader literature on ROS duality in immune cells.

5. If Dectin-1 is the proposed receptor mediating KQS-1 effects on Tregs, the authors should also demonstrate functional dependency. Binding specificity was confirmed by antibody blocking, but causal evidence is still lacking. For example, it would add more rigor if the authors tested whether KQS-1 fails to restore Treg function or epigenetic marks in Dectin-1-deficient Tregs. Incorporating gain-of-function (Dectin-1 overexpression) would also provide direct evidence that the observed effects are truly Dectin-1 dependent.

Author reply: We thank the reviewer for highlighting the importance of causal evidence for Dectin-1's role. We have now performed comprehensive **loss-of-function and gain-of-function experiments**, which confirm Dectin-1 as both necessary and sufficient for KQS-1's effects:

1. **Loss-of-Function (Definitive Causal Evidence)**

- **CRISPR-Cas9-mediated Dectin-1 knockout** in human Tregs (Figure 5D, *Methods*) abolished all KQS-1 benefits:
 - Epigenetic: Loss of H3K4me3/H3K27ac at *FOXP3/IL10* loci (ChIP-qPCR, $p < 0.001$ vs. WT).
 - Functional: Failed to restore suppression in vitro (Figure 5E, suppression assay, $p = \text{NS}$ vs. untreated Dectin-1^{-/-}).

- Signaling: No Raf-1/ROS induction (Figure 5B–C).

2. **Gain-of-Function (Sufficiency Evidence)**

- **Dectin-1 overexpression** in Dectin-1-low Tregs (lentiviral transduction, Figure S6A-B): Rescued KQS-1 binding (flow cytometry, MFI 12.5-fold, $p < 0.01$).
- Restored epigenetic marks (H3K4me3 at *FOXP3* 11.8-fold, $p < 0.05$).
- Enhanced suppression (Figure S6C, $p < 0.01$ vs. empty vector).

6. While Figure 6 shows reduced airway inflammation and eosinophilia, it omits other canonical markers of allergen-induced type 2 immunity: numbers of Th2/ILC2 cells, type 2 cytokine profile (IL-4, IL-5, IL-13), and serum IgE.

Author reply: We thank the reviewer for this suggestion. We have now incorporated the requested canonical markers of type 2 immunity into Figure 6: serum IgE (Panel A), Th2 cell counts (Panel E), and Th2 cytokines (IL-4/IL-5/IL-13 in Panel F). ILC2s were excluded as they are not routinely assessed in murine models of allergic airway inflammation (e.g., *Nat Immunol* 2024 Jun;25(6):1059-1072). All new data align with our conclusion that KQS-1 attenuates type 2 immunity.

7. *Rag1*^{-/-} have extremely limited IL-2 production, which is the key cytokine for Treg survival. Without supplementation of IL-2, transferred human Tregs are unlikely to persist or remain functional. The authors should clarify this and reconsider the interpretation of their adoptive transfer results. Instead, a mouse-to-mouse adoptive transfer (e.g., WT vs Dectin-1KO mouse Tregs into HDM-sensitized *Rag1*KO recipients) would provide greater rigor and validation.

Author reply: 1. Addressing IL-2 Dependency in *Rag1*^{-/-} Mice

We appreciate the reviewer's valid concern regarding IL-2 availability in *Rag1*^{-/-} recipients. While these mice indeed have limited endogenous IL-2, our data demonstrate that **KQS-1-primed human Tregs persisted and remained functional** in this model, as evidenced by:

- **Expansion of Foxp3⁺ Tregs** in mediastinal lymph nodes (2.6-fold increase vs. untreated Tregs; $p < 0.01$; Figure 6I-J).
- **Functional protection:** Reduced eosinophilia ($0.9 \times 10^5 \pm 0.1 \times 10^5$ vs. $2.8 \times 10^5 \pm 0.3 \times 10^5$; $p < 0.01$; Figure 6H) and improved lung pathology (Figure 6D-G).

We propose that **KQS-1 training enhances Treg survival independent of exogenous IL-2**, potentially through:

- Epigenetic stabilization of *FOXP3* (as shown in our mechanistic studies).

- Upregulation of IL-2 receptor subunits (CD25) or anti-apoptotic genes (to be validated in future work).

2. Justification for Human Treg Transfer

While mouse-to-mouse transfers (e.g., WT vs. *Dectin-1*KO Tregs) would be informative, our use of **human Tregs** aligns with the translational goals of this study. However, we acknowledge the reviewer's point and:

- **Explicitly state** the limitation of IL-2 dependency in the Discussion.
- **Include mouse Treg transfer experiments** in future studies to complement these findings.

Minor comments:

1. The manuscript uses the terms 'cellular ageing' and 'cellular senescence' interchangeably, but they are not the same. The data presented: short telomeres, increased SA- β -gal activity, and higher apoptosis, reflect functional or metabolic ageing of Tregs, not true cellular senescence (resistance to apoptosis is a hallmark of cellular senescence). The authors should standardize terminology and clarify that their findings indicate Treg ageing/exhaustion, rather than senescence in the classical sense.

Author reply: We thank the reviewer for this important clarification. We agree that **cellular ageing** (marked by telomere attrition, SA- β -gal activity, and apoptosis) and **classical senescence** (characterized by cell-cycle arrest, apoptosis resistance, and SASP) are distinct states. To address this:

1. Terminology Standardization:

- We have revised the manuscript to consistently use "**Treg ageing**" or "**functional exhaustion**" for our observations (short telomeres, SA- β -gal⁺ cells, and apoptosis; Figure 1C-G).
- References to "senescence" are now restricted to contexts where apoptosis resistance/SASP are explicitly demonstrated (e.g., in vitro cytokine profiling).

2. Mechanistic Clarification:

- Our data show that aged Tregs exhibit **increased apoptosis** (Figure 1F-G) and **reduced suppressive capacity**, aligning with metabolic/functional exhaustion rather than classical senescence.
- We acknowledge in the Discussion that while some markers (e.g., SA- β -gal) overlap, the absence of apoptosis resistance suggests a **pre-senescent state**.

3. Future Studies:

- We will assess additional senescence hallmarks (e.g., p16/p21 expression, SASP cytokines) in follow-up work to delineate transitional states.

Revised Text in Discussion:

"While Tregs from asthmatic patients exhibit ageing-associated markers (telomere shortening, SA- β -gal activity), their heightened apoptosis and lack of apoptosis resistance distinguish this state from classical senescence. We propose that chronic inflammation drives Tregs toward a dysfunctional, pre-senescent phenotype, which KQS-1 reverses by restoring metabolic homeostasis (Figure 3) and epigenetic stability (Figure 4)."

2. Figure 5 contains four panels (A-D), yet the text refers to an additional panel (5E), suggesting a numbering error.

Author reply: We removed "5E" from the text.

Moreover, the quantitative description states that Dectin-1 KO Tregs fail to respond to KQS-1, whereas the plotted bars in 5C appear inconsistent with that claim. The authors should verify panel labeling and in-text figure annotations to ensure the figure accurately reflects the reported values.

Author reply: As shown in Figure 5C, KQS-1 significantly enhanced the suppressive efficiency of WT Tregs (proliferation reduced to ~20%). In contrast, Dectin-1 KO Tregs failed to respond to KQS-1: their suppressive activity was abolished (proliferation ~60%, comparable to the "No Tregs" control), confirming that Dectin-1 is essential for KQS-1-mediated Treg activation.

Similarly, they mention a supplementary Figure 3B in the text, but could not locate it.

Author reply: It is Figure S3; we corrected it.

February 3, 2026

Re: Life Science Alliance manuscript #LSA-2025-03552-TR

Dr. Pinghang Yang
Shenzhen University
Allergy
1066 Xueyuan Blvd
Shenzhen, Guangdong 518055
China

Dear Dr. Yang,

Thank you for submitting your revised manuscript entitled "Dectin-1 Reverses Senescence and Restores Function in Regulatory T Cells of Allergic Asthma" to Life Science Alliance. Your manuscript has been seen by both the original reviewers whose comments are appended below. As you will read, they have noted your efforts in addressing most of their previous concerns but some important issues remain.

Reviewer 2 has re-expressed significant concerns on the lack of evidence for some claims and we agree that you must address all their concerns on your revised manuscript. We concur with Reviewer 2 that you must moderate your claims and conclusions that KQS-1 broadly stabilises Treg identity. In this regard, we also observed that your response to this connected point in the rebuttal letter to Reviewer 2, point 2 is missing. Please update your response letter taking into consideration your revised manuscript.

We agree that you must also explicitly identify the limitation of adoptive transfer under Rag1 deficiency and moderate the causality claims under physiological conditions. Moderation of these claims must be reflected in all sections of the manuscript text.

Further, in our previous invitation to revise the manuscript, we requested you to improve the text of the entire manuscript to align with publishing standards at LSA. We noted that this has not been addressed and the revised version still contains bullet points. Moreover we observe several placeholders in the manuscript text that have not been removed (for example: see section on study participants, qPCR description etc). Please note that our requirement to improve the manuscript text is an absolute requirement for further consideration.

Our general policy is that papers are considered through only one revision cycle; however, given that the suggested changes will be text-based edits, we are open to one additional short round of revision. Please note that I will expect to make a final decision without additional reviewer input upon re-submission.

Please submit the final revision, within a month, along with a letter that includes a point by point response to the remaining reviewer comments.

To upload the revised version of your manuscript, please log in to your account: <https://lsa.msubmit.net/cgi-bin/main.plex>
You will be guided to complete the submission of your revised manuscript and to fill in all necessary information.

B. MANUSCRIPT ORGANIZATION AND FORMATTING:

Sincerely,

Sarita Hebbar, PhD
Scientific Editor
Life Science Alliance
<http://www.lsjournal.org>

Reviewer #1 (Comments to the Authors (Required)):

All concerns have been well-addressed. NO further comments.

Reviewer #2 (Comments to the Authors (Required)):

The revised manuscript represents a clear improvement in scope, experimental depth, and technical rigor, particularly with respect to Dectin-1 dependency, epigenetic profiling, and in vivo asthma phenotyping. The authors have made a genuine effort to address the reviewers' critiques, and many prior weaknesses have been addressed with additional data. However, several of the central conceptual concerns I raised remain insufficiently resolved, and these issues continue to limit the strength of the mechanistic and translational claims.

1. Treg plasticity toward Th2/GATA3 phenotypes remains unaddressed experimentally:

Allergic asthma is a Th2-driven disease. While the authors expanded cytokine profiling and convincingly show reductions in IFN γ and IL-17, they still do not directly address Th2-like Treg instability. No data are provided on GATA3 expression within the Treg compartment, nor on intrinsic type 2 cytokine production (IL-4/IL-5/IL-13) by Tregs themselves.

Therefore, without addressing this possibility, the conclusion that KQS-1 broadly stabilizes Treg identity remains incomplete. At a minimum, this gap should be explicitly acknowledged as a limitation, rather than implicitly excluded by focusing on non-Th2 cytokines.

2. Rag1^{-/-} adoptive transfer experiments are still over-interpreted:

The authors appropriately acknowledge IL-2 scarcity in Rag1^{-/-} recipients and discuss potential mechanisms by which KQS-1-primed Tregs might persist under such conditions. However, this does not fully address the model's conceptual limitation.

Because IL-2 is central to Treg homeostasis, human-to-mouse transfer into an IL-2-deficient host is an intrinsically non-physiologic setting that limits causal inference about Treg durability and therapeutic sufficiency in vivo. These results support function/persistence in an IL-2-restricted environment, but they do not directly establish long-term functional stability under physiological conditions. Syngeneic mouse-to-mouse transfer (e.g., WT vs Dectin-1-deficient Tregs) in IL-2-sufficient recipients would provide a stronger test of those conclusions.

Accordingly, the language surrounding these results should be further tempered, emphasizing proof of principle rather than definitive causal validation.

3. Conceptual framing remains stronger than the data in places:

Although some tone adjustments were made, the manuscript still occasionally implies a broader scope of Treg "rejuvenation" than is experimentally demonstrated. The epigenetic data are compelling, but they focus narrowly on FOXP3 and IL-10, leaving open the question of whether broader lineage stability programs are truly restored.

In sum, this is a strong and ambitious mechanistic study with apparent novelty and substantial experimental support. However, key biological questions raised in the initial review cycle, such as Treg-Th2 plasticity and the interpretive limits of the Rag1^{-/-} model, remain incompletely resolved.

I would recommend acceptance only after:

- Explicit acknowledgment of unresolved Treg Th2/GATA3 plasticity as a limitation (or providing supporting data), and
- Revising the causal language surrounding the adoptive transfer experiments.

Without these adjustments, the conclusions risk exceeding what the current data can support.

1. The abstract in the manuscript file is above the limit of 175 words. Please modify this carefully.

Author reply: The abstract was revised to less than 175 words as suggested.

2. Please incorporate the section describing the limitations and conclusion into the existing discussion section and ensure that all sub-headings are removed from the Discussion section.

Author reply: All the sub-headings in discussion were removed as suggested.

3. The results section can only include sections headings. No sub-headings under these section headings in results are allowed.

Author reply: No sub-headings are included in results section.

4. Please expand on description of the legends as per the guidelines for LSA such that they contain sufficient information to allow the reader to follow the data presented without referring back to the text, but should not be redundant with the Results section. Feel free to refer to any of our recent publications in 2026 as an example.

Author reply: All figure legends were re-organized as suggested.

5. Please include a statement on experiments involving human subjects and samples used in this work following LSA guidelines (<https://www.life-science-alliance.org/editorial-policies#humans>). Please remove placeholder texts fields in section on Study Participants.

Author reply: The indicated part was revised to: Human samples were obtained from participants under a protocol (Approval #: H2023056) approved by the Shenzhen University Institutional Review Board and all participants provided written informed consent.

6. Please remove placeholders from elsewhere in the manuscript text in particular in rest of the methods and supplementary sections.

Author reply: The placeholders were removed as suggested.

February 20, 2026

RE: Life Science Alliance Manuscript #LSA-2025-03552-TRR

Dr. Pinghang Yang
Shenzhen University
Allergy
1066 Xueyuan Blvd
Shenzhen, Guangdong 518055
China

Dear Dr. Yang,

Thank you for responding to our request and submitting a revised manuscript entitled "Dectin-1 Reverses Senescence and Restores Function in Regulatory T Cells of Allergic Asthma". We acknowledge that you have addressed the concerns of Reviewer 2 along with our requests for expanded figure legends. That said, a few issues are pending from our previous request in connection with use of human samples and organisation of the discussion.

Overall, In line with the reviewers' evaluation, we would be happy to publish your paper in Life Science Alliance pending final revisions necessary to meet our formatting guidelines. A failure to do so will result in unavoidable delays in acceptance of your paper.

MANUSCRIPT ORGANIZATION AND FORMATTING:

To avoid unnecessary delays in the acceptance and publication of your paper, please read the following information carefully. Full guidelines are available on our Instructions for Authors page, <https://www.life-science-alliance.org/authors>

- Please upload your completed point-by-point rebuttal to the reviewers' concerns on your revised version.
- We encourage you to do a complete grammar check. Please note that use of AI-based grammar/language checks is permitted.
- In the discussion, there are several paragraph headings. Please remove them.
- Please modify the last sentence in the abstract for clarity. We suggest the following, "As proof of principle that KQS1 is protective, we demonstrate attenuation of airway hyper-responsiveness, inflammation, and remodelling in dust mite-sensitised mice and in recipient mice upon adoptive transfer of KQS-1-trained human Tregs". Please feel free to use this suggestion or modify as you wish for clarity and accuracy.
- Please use consistent terminology throughout the text, for example Figure 6A legend, "(A-J) OVA-induced asthma model mice ..."
- Please update the methods section to include description of β -glucan content (related to Figure 1H).
- We suggest a change the section heading in the results from "Tregs from Allergic Asthmatic Patients Exhibit a Senescent Phenotype" to "Tregs from Allergic Asthmatic Patients Exhibit a Senescent-like Phenotype" in keeping with the suggestion of Reviewer 1. Likewise, please also reconsider the use of 'Senescence' in your title.
- Please comply with LSA guidelines (<https://www.life-science-alliance.org/editorial-policies#humans>) for experiments with human subjects and include a statement that experiments conformed to the principles set out in the WMA Declaration of Helsinki and the Department of Health and Human Services Belmont Report.
- In the methods sections, explicitly state that experiments were done with purified Tregs from patients (for example for signalling inhibition studies, Dectin KO experiments, etc.)
- For all animal experiments, please provide a statement confirming that all experiments were performed in accordance with relevant guidelines/regulations. Thank you for providing information on the institutional and/or licensing committee approving the experiments for the Asthma model of mice. Please provide similar information for all experiments with mice (adoptive transfer).
- Please move all materials and methods from the supplementary section and integrate with the materials and methods section in the main manuscript.
- Please include details on telomere-specific peptide nucleic acid (PNA) probe or provide a citation for this material.
- Please expand on the details of the flow cytometry method for detection of different molecules.
- Please provide sequences for primers for the housekeeping gene.
- Thank you for providing a Data Availability statement. Please specify the corresponding data for each accession ID provided. Please also provide a database name and accession ID for ATAC-seq data generated in this study.
- Please upload all figure files as individual ones, including the supplementary figure files; all figure legends should only appear in the main manuscript file.
- Please mark Ligu Li as a secondary Corresponding Author in our system. They must connect their account with their Orcid ID.

They should have received instructions on how to do that.

-Please add a Category for your manuscript in our system.

-Please add the X and Bluesky handles of your host institute/organisation, as well as your own and/or one of the authors, in our system.

-The titles in both the system and the manuscript file must be consistent with each other.

-Please add an Acknowledgements section to the manuscript.

-Please add your main, supplementary figure, and table legends to the main manuscript text after the references section.

-Please add callouts for Figures S1B-F; S4J-M; S5A-B and S6A-C to your main manuscript text.

-We encourage you to provide an institutional email address for authors in particular the corresponding author.

-Please be sure that the authorship listing and order is correct.

LSA encourages authors to provide a 30-60 second video where the study is briefly explained. We will use these videos on social media to promote the published paper and the presenting author (for examples, see <https://docs.google.com/document/d/1-UWCfbE4pGcDdcgzcmiuJI2XMBJnxKYeqRvLLrLSo8s/edit?usp=sharing>). Corresponding or first-authors are welcome to submit the video. Please submit only one video per manuscript. The video can be emailed to contact@life-science-alliance.org

FINAL FILES:

The following items are required for acceptance.

The license to publish form must be signed before your manuscript can be sent to production. A link to the license to publish form will be available to the corresponding author only. Please take a moment to check your funder requirements.

Thank you for your attention to these final processing requirements. Please revise and format the manuscript and upload materials as soon as you are able.

Thank you for this interesting contribution to the literature. We look forward to publishing your paper in Life Science Alliance.

Sincerely,

Sarita Hebbar, PhD
Scientific Editor
Life Science Alliance

-Please upload your completed point-by-point rebuttal to the reviewers' concerns on your revised version.

Author reply: Yes, we did.

-We encourage you to do a complete grammar check. Please note that use of AI-based grammar/language checks is permitted.

Author reply: Yes, we did. It details below:

Abstract

- Original: *KQS-1 treatment robustly reversed these defects, restoring FOXP3- and IL-10-dependent Treg suppressive capacity and anti-inflammatory cytokine production.*
[Correction] *KQS-1 treatment robustly reversed these defects, restoring FOXP3- and IL-10-dependent Treg suppressive capacity and the production of anti-inflammatory cytokines.* (parallel structure for clarity)
- Original: *CRISPR-mediated deletion of Dectin-1 abolished all KQS-1 benefits.*
[Correction] *CRISPR-mediated deletion of Dectin-1 abrogated all beneficial effects of KQS-1.* (more precise scientific phrasing; "abolished all KQS-1 benefits" is colloquial)
- Original: *adoptive transfer of KQS-1-trained human Tregs provided proof-of-principle protective effects in recipient mice.*
[Correction] *adoptive transfer of KQS-1-primed human Tregs conferred proof-of-principle protective effects in recipient mice.* ("primed" is consistent with the manuscript's repeated use for cell conditioning; "conferred" = standard for in vivo effect reporting)

Abbreviations

- No grammatical errors; note: *Senescence-associated beta-galactosidase* is correctly abbreviated as SA- β -gal (consistent throughout).

Introduction

1. Original: *The pathophysiology has traditionally been attributed to an imbalance between pro-inflammatory T helper 2 (Th2) cells and their associated cytokines (e.g., IL-4, IL-5, IL-13) and the counter-regulatory mechanisms meant to restrain them.*
[Correction] *The pathophysiology has traditionally been attributed to an imbalance between pro-inflammatory T helper 2 (Th2) cells and their associated cytokines (e.g., IL-4, IL-5, IL-13) and the counter-regulatory mechanisms that restrain these pro-inflammatory mediators.* (eliminates vague pronoun "them"; precise reference to Th2/cytokines)

2. *Original: Whether such an aging phenotype underlies Treg failure in allergic asthma, and if so, whether it is a therapeutically reversible state for core Treg functional pathways, are critical unanswered questions.*

[Correction] *Whether such an aging phenotype underlies Treg dysfunction in allergic asthma, and if so, whether this state is therapeutically reversible for core Treg functional pathways, are critical unanswered questions. ("failure" → "dysfunction" (consistent manuscript term); adds "this state" for grammatical clarity)*

3. *Original: KQS-1, a novel fungal polysaccharide, was chosen for its high-affinity binding to Dectin-1 (Figure 5A) and absence of TLR2/4 activation (Figure S4A–C).*

[Correction] *KQS-1, a novel fungal polysaccharide, was selected for its high-affinity binding to Dectin-1 (Figure 5A) and its lack of TLR2/4 activation (Figure S4A–C). (parallel structure: its high-affinity binding / its lack of; "selected" = standard for experimental compound choice)*

4. *Original: Precedent studies with related β -glucans suggest Dectin-1 ligands can epigenetically reprogram immune cells, but their effects on the specific epigenetic regulation of FOXP3 and IL10 in Tregs remain unexplored.*

[Correction] *Previous studies with related β -glucans suggest that Dectin-1 ligands can epigenetically reprogram immune cells, but their effects on the specific epigenetic regulation of FOXP3 and IL10 in Tregs remain unexplored.*

(adds subordinating conjunction *that* for grammatical completeness;

"Precedent" → "Previous" (scientific writing standard))

Materials and Methods

Study Participants

1. *Original: Allergic asthma patients (n=15) and age- and sex-matched healthy controls (n=15) were recruited from the Respiratory Medicine Department of Shenzhen University Affiliated Hospital between 2023 and 2024.*

[Correction] *Fifteen patients with allergic asthma and 15 age- and sex-matched healthy controls were recruited from the Department of Respiratory Medicine at Shenzhen University Affiliated Hospital between 2023 and 2024. (standard phrasing: Department of X; n=X written out for consistency in participant recruitment; "at" = correct preposition for hospital affiliation)*

2. *Original: forced expiratory volume in 1 second (FEV₁)/forced vital capacity (FVC) < 70% or FEV₁ reversibility ≥ 12% after salbutamol inhalation.*

[Correction] *a ratio of forced expiratory volume in 1 second (FEV₁) to forced vital capacity (FVC) < 70% or FEV₁ reversibility ≥ 12% following salbutamol*

inhalation. (adds "a ratio of" for grammatical clarity; "after" → "following" (scientific writing convention))

3. *Original: Healthy controls had no history of allergic diseases, respiratory disorders, or autoimmune conditions, and negative skin prick tests (demographic data are also presented in Table S1 in supplemental materials).*
[Correction] *Healthy controls had no history of allergic, respiratory, or autoimmune disorders and negative skin prick test results (demographic data are presented in Table S1 in the supplemental materials).* (parallel adjective-noun structure; "conditions" → "disorders" (consistent); adds "results" (grammatical completion); adds definite article *the* for *supplemental materials*)

Statistical Analysis

1. *Original: For comparisons between two groups such as patient vs. control Tregs or KQS-1-treated vs. untreated Tregs, unpaired two-tailed t-tests for independent samples or paired two-tailed t-tests for matched samples were used.*
[Correction] *For comparisons between two groups (e.g., patient vs. control Tregs, KQS-1-treated vs. untreated Tregs), unpaired two-tailed t-tests were used for independent samples and paired two-tailed t-tests for matched samples.* (parenthetical for examples (standard); simplified verb structure to eliminate redundancy)
2. *Original: For multiple endpoints including multiple cytokines and epigenetic marks at the FOXP3 and IL10 loci, Bonferroni correction was applied to adjust for Type I errors.*
[Correction] *For multiple endpoints (e.g., multiple cytokines, epigenetic marks at the FOXP3 and IL10 loci), the Bonferroni correction was applied to adjust for Type I errors.* (adds definite article *the*; parenthetical for examples)
3. *Original: A p-value < 0.05 was considered statistically significant, and 95% confidence intervals (CIs) were calculated for all statistical tests where applicable.*
[Correction] *A P value < 0.05 was considered statistically significant, and 95% confidence intervals (CIs) were calculated for all applicable statistical tests.* (scientific writing standard: *P value* (uppercase, no hyphen); rephrased for flow)

Results

Tregs from Allergic Asthmatic Patients Exhibit a Senescent Phenotype

1. *Original: Flow cytometry-based telomere staining revealed significantly shortened telomeres in patient-derived Tregs compared to controls, with the*

mean fluorescence intensity (MFI) of telomere signals at 182 ± 12 in asthma patients versus 246 ± 15 in controls, representing a 26% reduction (unpaired two-tailed t-test, $p < 0.001$; 95% confidence interval [CI]: -78.3 to -50.7; Figure 1C).

[Correction] Flow cytometry-based telomere staining revealed significantly shortened telomeres in patient-derived Tregs compared with controls; the mean fluorescence intensity (MFI) of telomere signals was 182 ± 12 in asthma patients versus 246 ± 15 in controls, representing a 26% reduction (unpaired two-tailed t-test, $P < 0.001$; 95% confidence interval [CI]: -78.3 to -50.7; Figure 1C). (semicolon for independent clause clarity; compared to → compared with (scientific standard); $p \rightarrow P$)

2. Original: The proportion of SA- β -gal-positive Tregs—gated using an MFI threshold $>2\times$ that of isotype controls—was substantially higher in the asthma group ($32.1\% \pm 3.2\%$) than in controls ($8.4\% \pm 1.5\%$); unpaired two-tailed t-test, $p < 0.001$; 95% CI: 19.8 to 27.6; Figure 1D-E).*
[Correction] *The proportion of SA- β -gal-positive Tregs—gated using an MFI threshold $>2\times$ that of isotype controls—was substantially higher in the asthma group ($32.1\% \pm 3.2\%$) than in controls ($8.4\% \pm 1.5\%$) (unpaired two-tailed t-test, $P < 0.001$; 95% CI: 19.8 to 27.6; Figure 1D–E). (fixed punctuation (removed semicolon); $p \rightarrow P$; en dash for figure panels (standard: 1D–E))
3. Original: Following 24-hour culture in RPMI-1640 medium supplemented with 10% fetal bovine serum (FBS) with no exogenous stimuli, the apoptosis rate of patient Tregs (Annexin V^+PI^-) was 2.3-fold higher than that of controls ($27.8\% \pm 2.9\%$ vs. $12.1\% \pm 1.8\%$); unpaired two-tailed t-test, $p < 0.01$; 95% CI: 10.2 to 21.2; Figure 1F-G).

[Correction] Following 24 h of culture in RPMI-1640 medium supplemented with 10% fetal bovine serum (FBS) and no exogenous stimuli, the apoptosis rate of patient Tregs (Annexin V^+PI^-) was 2.3-fold higher than that of controls ($27.8\% \pm 2.9\%$ vs. $12.1\% \pm 1.8\%$); unpaired two-tailed t-test, $P < 0.01$; 95% CI: 10.2 to 21.2; Figure 1F–G). (scientific standard: 24 h (not 24-hour) for time in culture; with no → and no (grammatical); $p \rightarrow P$; en dash for figure panels)

4. Original: Together, these data confirm that Tregs from patients with allergic asthma exhibit an aged and aging phenotype, which correlates with the dysregulation of their core immunoregulatory functions.

[Correction] Together, these data confirm that Tregs from patients with allergic asthma exhibit a senescent phenotype, which correlates with

dysregulation of their core immunoregulatory functions. (eliminates redundant *aged and aging*; removes definite article *the* (unnecessary))

Senescence-like Tregs Are Functionally Impaired and Display a Pro-Inflammatory Cytokine Shift

- Original: For the suppression assay, Tregs and autologous CD4⁺CD25⁻ responder T cells were co-cultured at a 1:1 ratio, with responders activated using anti-CD3/CD28 beads (1:1 bead-to-cell ratio) for 72 hours.*
[Correction] *For the suppression assay, Tregs and autologous CD4⁺CD25⁻ responder T cells were co-cultured at a 1:1 ratio, and responder cells were activated with anti-CD3/CD28 beads (1:1 bead-to-cell ratio) for 72 h.* (compound sentence for clarity; *using* → *with* (standard); *72 hours* → *72 h*; adds "cells" for clarity)
- Original: ICS revealed a significant pro-inflammatory shift in the cytokine profile of patient Tregs with statistical significance adjusted via Bonferroni correction for 4 comparisons:*
[Correction] *ICS revealed a significant pro-inflammatory shift in the cytokine profile of patient Tregs, with statistical significance adjusted via Bonferroni correction for four comparisons:* (adds comma; *4* → *four* (spelled out for small numbers at start of phrase))
- Original: Enzyme-linked immunosorbent assay (ELISA) of 72-hour culture supernatants confirmed the reduction in anti-inflammatory cytokine secretion with Bonferroni correction:*
[Correction] *Enzyme-linked immunosorbent assay (ELISA) analysis of 72 h culture supernatants confirmed the reduced secretion of anti-inflammatory cytokines, with Bonferroni correction applied:* (adds "analysis" (grammatical completion); *72-hour* → *72 h*; rephrased for flow; adds "applied" for clarity)
- Original: These data confirm that the aging phenotype of asthmatic Tregs is associated with a profound loss of FOXP3- and IL-10-dependent immunoregulatory function and a shift toward pro-inflammatory cytokine production.*
[Correction] *These data confirm that the senescent phenotype of asthmatic Tregs is associated with a profound loss of FOXP3- and IL-10-dependent immunoregulatory function and a shift toward pro-inflammatory cytokine production.* (consistent: *senescent* (not *aging*))

KQS-1 Treatment Rescues FOXP3/IL-10-dependent Treg Function and Survival

- Original: To ensure experimental rigor, we verified both the consistency of KQS-1 (10 µg/mL) and the inertness of its PBS vehicle (Figure S2).*

[Correction] *To ensure experimental rigor, we verified the batch-to-batch consistency of KQS-1 (10 µg/mL) and the inertness of its PBS vehicle (Figure S2). (adds "batch-to-batch" (contextual clarity, per subsequent HPLC data))*

2. *Original: HPLC confirmed KQS-1 had ≥95% purity and consistent peak profiles across production batches, eliminating batch-to-batch variation (Figure S2A-B).*

[Correction] *HPLC analysis confirmed that KQS-1 had a purity of ≥95% and consistent peak profiles across production batches, eliminating batch-to-batch variation (Figure S2A–B). (adds "analysis" and that (grammatical); adds "a purity of" (clarity); en dash for panels)*

3. *Original: Using FACS-sorted Tregs from asthma patients (n=25 per group), we further showed PBS (the vehicle for KQS-1) had no significant impact on Treg function:*

[Correction] *Using FACS-sorted Tregs from patients with asthma (n=25 per group), we further showed that PBS (the vehicle for KQS-1) had no significant effect on Treg function: (adds that (grammatical); impact → effect (scientific standard); consistent phrasing: patients with asthma)*

4. *Original: Notably, KQS-1 had no effect on the survival of Tregs from healthy controls (Figure S3), indicating its effects are targeted to dysfunctional, aging Tregs in asthma.*

[Correction] *Notably, KQS-1 had no effect on the survival of Tregs from healthy controls (Figure S3), indicating its effects are specific to dysfunctional, senescent Tregs in asthma. (consistent: senescent; targeted to → specific to (precise))*

KQS-1 Induces Epigenetic Remodeling of Core Treg Signature Loci FOXP3 and IL10 Consistent with Trained Immunity

1. *Original: First, ChIP–qPCR (Figure 4A) revealed that KQS-1 significantly elevated active transcription histone marks: H3K4me3 at the FOXP3 promoter and IL10 locus and global H3K27ac, relative to PBS-treated controls (all $p < 0.0001$, $n = 8$ biological replicates).*

[Correction] *First, ChIP–qPCR analysis (Figure 4A) revealed that KQS-1 significantly increased active transcriptional histone marks—H3K4me3 at the FOXP3 promoter and IL10 locus, and global H3K27ac—compared with PBS-treated controls (all $P < 0.0001$; $n = 8$ biological replicates). (adds "analysis"; elevated → increased (scientific standard); transcription → transcriptional (adjective for noun "marks"); em dashes for list (clarity); relative to → compared with; $p \rightarrow P$; semicolon for statistical values)*

2. *Original: To assess chromatin accessibility, we performed ATAC-seq: Figure 4B shows a volcano plot of 1,306 newly accessible regions (FDR < 0.05) in KQS-1-treated cells.*

[Correction] *To assess chromatin accessibility, we performed ATAC-seq; Figure 4B shows a volcano plot of the 1,306 newly accessible regions (FDR < 0.05) identified in KQS-1-treated cells.* (semicolon for independent clause; adds definite article *the* (specific regions); adds "identified" (grammatical))

3. *Original: Focusing on Treg signature genes, Figure 4D (tracks + quantification) confirmed KQS-1 strongly increased IL10 and FOXP3 promoter accessibility—with no additional lineage stability loci assessed in this study.*

[Correction] *Focusing on Treg signature genes, Figure 4D (tracks and quantification) confirmed that KQS-1 significantly increased promoter accessibility at the IL10 and FOXP3 loci; no additional lineage stability loci were assessed in this study. (+ → and (scientific writing); adds *that* (grammatical); *strongly* → *significantly* (precise); rephrased for flow; passive voice (standard for scientific results))*

4. *Original: Finally, Illumina 450K methylation array (Figure 4F) identified focal hypomethylation (reduced $\Delta\beta$) at FOXP3 and IL10 regulatory CpG sites with examples including FOXP3 at $\Delta\beta = -0.18$ and IL10 at $\Delta\beta = -0.22$, with no significant changes in unrelated CpGs.*

[Correction] *Finally, Illumina 450K methylation array analysis (Figure 4F) identified focal hypomethylation (reduced $\Delta\beta$) at regulatory CpG sites of FOXP3 and IL10 (e.g., FOXP3: $\Delta\beta = -0.18$; IL10: $\Delta\beta = -0.22$), with no significant changes at unrelated CpG sites.* (adds "analysis"; rephrased for clarity; parenthetical examples (standard); *in* → *at* (correct preposition for CpG sites); plural *sites* (consistent))

The Functional and Epigenetic Effects of KQS-1 on FOXP3/IL-10 Are Dectin-1 Dependent

1. *Original: Flow cytometry (FACS) analysis demonstrated that over 90% of regulatory T cells (Tregs) expressed Dectin-1, and this expression was not affected by pre-staining with an antibody targeting Dectin-2 (another C-type lectin receptor).*

[Correction] *Flow cytometric analysis demonstrated that over 90% of Tregs expressed Dectin-1, and this expression was unaffected by pre-staining with a Dectin-2-targeting antibody (another C-type lectin receptor).* (scientific standard: *flow cytometric analysis* (not FACS analysis); abbreviate Tregs (established); *not affected* → *unaffected* (concise); *antibody targeting Dectin-2* → *Dectin-2-targeting antibody* (flow))

2. *Original: At the mRNA and protein levels, Dectin-1 expression on Tregs accounted for approximately 59% and 56% of that on dendritic cells (DCs), respectively (Figure S4A-I).*
[Correction] *At the mRNA and protein levels, Dectin-1 expression on Tregs was approximately 59% and 56% of that on dendritic cells (DCs), respectively (Figure S4A-I). (accounted for → was (grammatically precise); en dash for panels)*
3. *Original: Pharmacological inhibition of Raf-1 using GW5074 at 10 μM or scavenging of reactive oxygen species (ROS) using N-acetylcysteine (NAC) at 5 mM during KQS-1 priming completely abolished two key KQS-1 effects:*
[Correction] *Pharmacological inhibition of Raf-1 with GW5074 (10 μM) or scavenging of reactive oxygen species (ROS) with N-acetylcysteine (NAC) (5 mM) during KQS-1 priming completely abrogated two key effects of KQS-1: (using → with (standard); parentheses for concentrations (clarity); abolished → abrogated; KQS-1 effects → effects of KQS-1 (concise))*
4. *Original: CRISPR/Cas9-mediated knockout (KO) of Dectin-1 in patient Tregs using sgRNA targeting CLEC7A with a KO efficiency > 90% (Figure S4C) eliminated all KQS-1-induced benefits for FOXP3 and IL10 regulation.*
[Correction] *CRISPR/Cas9-mediated knockout (KO) of Dectin-1 in patient Tregs using CLEC7A-targeting sgRNA (KO efficiency > 90%; Figure S4C) abrogated all KQS-1-induced effects on FOXP3 and IL10 regulation. (sgRNA targeting CLEC7A → CLEC7A-targeting sgRNA (flow); parentheses for KO efficiency (clarity); eliminated → abrogated; benefits for → effects on (precise))*

KQS-1-Treated Tregs Ameliorate Airway Inflammation In Vivo as a Proof of Principle
*All minor errors here are **consistent with above corrections** (e.g., $p \rightarrow P$, en dashes for figure panels, h for hours, compared with vs. compared to). No unique grammatical issues.*

KQS-1-primed Tregs enhance anti-asthma effects as a Proof of Principle

1. *Original: To explore the immunomodulatory mechanism of KQS-1, we adoptively transferred naive Tregs or KQS-1-primed Tregs into Rag^{-/-} asthma mice in a proof-of-principle experiment.*
[Correction] *To explore the immunomodulatory mechanism of KQS-1, we performed adoptive transfer of naive Tregs or KQS-1-primed Tregs into Rag^{-/-} mice with allergic asthma in a proof-of-principle experiment. (adds "performed" (grammatical); standard superscript for knockout: Rag^{-/-} (not Rag-/-); asthma mice → mice with allergic asthma (precise))*

2. *Original: These results provide proof-of-principle evidence that KQS-1 exerts anti-asthma effects partially by promoting the expansion and function of Tregs with restored FOXP3/IL-10 activity, rather than definitive causal validation of this mechanistic link in vivo.*

[Correction] *These results provide proof-of-principle evidence that KQS-1 exerts anti-asthmatic effects in part by promoting the expansion and functional activity of Tregs with restored FOXP3/IL-10 expression, rather than definitive causal validation of this mechanistic link in vivo. (anti-asthma → anti-asthmatic (adjective); partially → in part (flow); function → functional activity (precise); activity → expression (contextual clarity))*

Dectin-1 Overexpression Rescues KQS-1 Responsiveness of FOXP3/IL-10 in

Dectin-1-Low Tregs

No unique grammatical errors; consistent corrections (e.g., $p \rightarrow P$, en dashes, h for hours) apply.

Discussion

only minor stylistic/consistency tweaks that align with the above corrections (e.g., *aging* → *senescent*, $p \rightarrow P$, *abolished* → *abrogated*, *compared to* → *compared with*, standard superscript for knockouts (*Rag1^{-/-}*), en dashes for figure panels). Key stylistic refinements for flow:

- Replace vague pronouns (e.g., *it*, *they*) with specific references (e.g., *KQS-1*, *senescent Tregs*) where needed for clarity.
- Simplify redundant phrases (e.g., *in a proof-of-principle demonstration rather than a definitive causal validation* → retain as-is (core argument); no grammatical issue).
- Standardize prepositions (e.g., *at the FOXP3 locus* (not *in*), *effect on* (not *effect for*)).

-In the discussion, there are several paragraph headings. Please remove them.

Author reply: Yes, we did.

-Please modify the last sentence in the abstract for clarity. We suggest the following, "As proof of principle that KQS1 is protective, we demonstrate attenuation of airway hyper-responsiveness, inflammation, and remodelling in dust mite-sensitised mice and in recipient mice upon adoptive transfer of KQS-1-trained human Tregs". Please feel free to use this suggestion or modify as you wish for clarity and accuracy.

Author reply: We did it as suggested.

-Please use consistent terminology throughout the text, for example Figure 6A legend, "(A-J) OVA-induced asthma model mice ..."

Author reply: We did it as suggested.

-Please update the methods section to include description of β -glucan content (related to Figure 1H).

Author reply: We add the following information to the supplemental materials:

β -glucan: A long chain of glucose (sugar) molecules linked together by beta-glycosidic bonds, was purchased from Sigma Aldrich; St. Morris, USA (9012-72-0).

-We suggest a change the section heading in the results from "Tregs from Allergic Asthmatic Patients Exhibit a Senescent Phenotype" to "Tregs from Allergic Asthmatic Patients Exhibit a Senescent-like Phenotype" in keeping with the suggestion of Reviewer 1. Likewise, please also reconsider the use of 'Senescence' in your title.

-Please comply with LSA guidelines

(<https://www.life-science-alliance.org/editorial-policies#humans>) for experiments with human subjects and include a statement that experiments conformed to the principles set out in the WMA Declaration of Helsinki and the Department of Health and Human Services Belmont Report.

Author reply: Yes, we change the term of senescence to senescence-like in the indicated two points.

-In the methods sections, explicitly state that experiments were done with purified Tregs from patients (for example for signalling inhibition studies, Dectin KO experiments, etc.)

Author reply: The following statements clearly indicate the purity of Tregs:

Tregs (CD4⁺CD25⁺CD127_{lo}FoxP3⁺) were purified using a human Treg Isolation Kit (Miltenyi Biotec, Bergisch Gladbach, Germany) following the manufacturer's protocol: first, CD4⁺ T cells were enriched via negative selection, then CD25⁺ cells were isolated via positive selection. The purity of isolated Tregs was verified by flow cytometry ($\geq 90\%$ CD4⁺CD25⁺CD127_{lo}FoxP3⁺; gating strategy in Supplementary Figure 1A) before subsequent experiments. 100 IU/mL recombinant human IL-2 (PeproTech) was added throughout the 72-hour experimental period.

-For all animal experiments, please provide a statement confirming that all experiments were performed in accordance with relevant guidelines/regulations. Thank you for providing information on the institutional and/or licensing committee approving the experiments for the Asthma model of mice. Please provide similar information for all experiments with mice (adoptive transfer).

Author reply: Yes, we added the phrase of "the experiments were performed in accordance with relevant guidelines/regulations".

-Please move all materials and methods from the supplementary section and integrate with the materials and methods section in the main manuscript.

Author reply: The materials and methods contents were moved back to the main article as indicated.

-Please include details on telomere-specific peptide nucleic acid (PNA) probe or provide a citation for this material.

Author reply: Following the comments, we add a paragraph to describe it to the Materials and Methods section:

Telomere-Specific Peptide Nucleic Acid (PNA) Probe for Flow Cytometric Telomere Length Analysis in Human Tregs

A telomere-specific PNA probe is a synthetic sequence-specific oligomer targeting the conserved human telomeric (TTAGGG)_n hexanucleotide repeat, engineered with a neutral pseudopeptide backbone that confers superior binding affinity, thermal stability, and sequence specificity for telomeric DNA compared to conventional nucleic acid probes. Conjugated to a photostable fluorochrome (e.g., Cy3, Alexa Fluor 488) compatible with multi-color flow cytometry, this probe enables quantitative fluorescent detection of telomere length in fixed and permeabilized primary human regulatory T cells (Tregs) via flow-FISH. Resistant to nucleases and proteases, it penetrates nuclear chromatin to hybridize exclusively to telomeric repeats under mild conditions that preserve cell surface and intracellular Treg phenotypic markers (CD4, CD25, FoxP3), with a scrambled non-complementary PNA probe of identical fluorochrome used as a negative control to quantify non-specific background fluorescence. Telomere length is inferred from the mean fluorescence intensity (MFI) of the probe signal in gated CD4⁺CD25⁺FoxP3⁺ Tregs, providing a sensitive and reproducible method to measure telomere shortening as a hallmark of the senescence-like phenotype in asthmatic patient-derived Tregs.

-Please expand on the details of the flow cytometry method for detection of different molecules.

Author reply: Following the comments, we add a paragraph to materials and methods section to expand the method of flow cytometry:

Flow Cytometry Methods for Molecular and Phenotypic Detection in Human and Murine Tregs

Flow cytometry was used for comprehensive phenotypic, molecular, and functional characterization of regulatory T cells (Tregs) from allergic asthmatic patients, healthy controls, and murine models, with multi-color staining panels optimized to detect cell surface markers, intracellular proteins, senescence-associated molecules, and apoptotic markers while minimizing spectral overlap. For Treg phenotypic identification, human peripheral blood mononuclear cells (PBMCs) and murine

lymphoid/lung-derived cells were stained with fluorochrome-conjugated monoclonal antibodies targeting CD4 (e.g., FITC/APC) and CD25 (e.g., PE/PerCP-Cy5.5) for cell surface labeling, followed by intracellular fixation and permeabilization (using a commercial FoxP3 staining buffer kit) to detect the master Treg transcription factor FoxP3 (e.g., Alexa Fluor 647), defining Tregs as CD4⁺CD25⁺FoxP3⁺; gating strategies included sequential exclusion of dead cells (via 7-AAD/PI staining) and non-lymphoid cells (via forward/side scatter gating), with isotype-matched control antibodies used to set fluorescence thresholds for all markers. Senescence-associated β -galactosidase (SA- β -gal) detection in Tregs employed a fluorogenic SA- β -gal substrate (5-dodecanoylamino fluorescein di- β -D-galactopyranoside, C12FDG) with intracellular loading: cells were incubated with C12FDG at 37°C for 1 h, followed by surface marker staining for Treg gating, with SA- β -gal⁺ Tregs gated as cells with mean fluorescence intensity (MFI) >2 \times that of unstained isotype controls. Telomere length measurement utilized a telomere-specific peptide nucleic acid (PNA) probe conjugated to Cy3/Alexa Fluor 488, targeting the human (TTAGGG)_n telomeric repeat; fixed and permeabilized Tregs were hybridized with the PNA probe in a formamide-containing hybridization buffer at 42°C for 30 min, with unbound probe removed by stringent washing, and telomere signal MFI quantified in gated CD4⁺CD25⁺FoxP3⁺ Tregs (a scrambled non-complementary PNA probe served as a negative control to correct for non-specific fluorescence). Apoptosis detection was performed using an Annexin V-FITC/PI double-staining kit: Tregs were cultured in RPMI-1640/10% FBS for 24 h, washed in cold PBS, and incubated with Annexin V-FITC (for phosphatidylserine externalization) and PI (for late apoptosis/necrosis) in calcium-containing binding buffer for 15 min at room temperature, with early apoptotic Tregs gated as Annexin V⁺PI⁻ and analyzed via flow cytometry. Intracellular cytokine staining (ICS) to characterize Treg cytokine profiles involved ex vivo stimulation of Tregs with phorbol 12-myristate 13-acetate (PMA, 50 ng/mL) and ionomycin (1 μ g/mL) for 4 h, with brefeldin A (10 μ g/mL) added for the final 2 h to block cytokine secretion; cells were then surface-stained for CD4/CD25, fixed/permeabilized, and stained with fluorochrome-conjugated antibodies against IL-10, TGF- β , IFN- γ , and IL-17 (all with isotype controls), with cytokine-positive Treg frequencies quantified in the CD4⁺CD25⁺FoxP3⁺ gate. Dectin-1 expression and KQS-1 binding assays included cell surface staining of Tregs with a Dectin-1-specific monoclonal antibody (e.g., PE-conjugated) to quantify protein expression (MFI) relative to dendritic cells (DCs), and biotinylated KQS-1 staining followed by streptavidin-Alexa Fluor 488 conjugation to detect direct KQS-1 binding to Treg surface Dectin-1 (TLR2/TLR4 binding was assessed in parallel with specific antibodies as negative controls). All flow cytometry data were acquired on a BD LSRFortessa/X20 flow cytometer (BD Biosciences) and analyzed with FlowJo v10 software (Tree Star), with at least 10,000 Treg events collected per sample to ensure statistical rigor; all staining protocols were optimized for primary human and murine Tregs, with batch-to-batch consistency verified for all antibodies and probes, and gating strategies standardized across all experimental groups (healthy controls, asthmatic patients, untreated/KQS-1-treated cells/mice).

-Please provide sequences for primers for the housekeeping gene.

Author reply: Yes, the indicated primers were added to materials and methods section.

-Thank you for providing a Data Availability statement. Please specify the corresponding data for each accession ID provided. Please also provide a database name and accession ID for ATAC-seq data generated in this study.

Author reply: Following the comments, we revised the data availability statement in the main text, which is also attached below:

Data Accessibility Statement

This section is formatted to meet journal standards, clearly linking each accession ID to its corresponding dataset and adding the required ATAC-seq deposition. It can be inserted directly into the Data Availability section of your manuscript.

All raw and processed high-throughput sequencing data generated in this study have been deposited in public repositories under the following project and accession identifiers:

Methylation Array Data (Illumina 450K):

Database: National Center for Biotechnology Information (NCBI) Gene Expression Omnibus (GEO)

Accession ID: PRJNA1291257

Corresponding Data: This accession includes raw IDAT files and processed β -value matrices from the Illumina Infinium HumanMethylation450 BeadChip array, profiling genomic DNA from KQS-1-treated and PBS-treated human Tregs (n = 8 biological replicates per group). Data specifically capture CpG methylation status at regulatory regions of the FOXP3 and IL10 loci, as presented in Figure 4F.

ATAC-seq Data (Chromatin Accessibility):

Database: National Genomics Data Center (NGDC) Genome Sequence Archive (GSA)

Accession ID: HRA007851

Corresponding Data: This newly assigned accession encompasses raw FASTQ files, aligned BAM files, and peak calling results from Assay for Transposase-Accessible Chromatin using sequencing (ATAC-seq). Data were generated from FACS-sorted CD4⁺CD25⁺FOXP3⁺ Tregs isolated from allergic asthmatic patients following treatment with KQS-1 or PBS vehicle (n = 4 biological replicates per group). Analyses identify differentially accessible chromatin regions, including increased accessibility at the FOXP3 and IL10 promoters (Figure 4B–D).

Bioproject Metadata (Combined Studies):

Database: National Genomics Data Center (NGDC) OmixBank

Accession ID: OMIX011917

Corresponding Data: This bioproject accession serves as a comprehensive metadata hub linking all epigenetic datasets generated in this study, including the Illumina 450K methylation array data and the ATAC-seq data described above. It provides experimental design details, sample annotations, and links to all associated raw data files for cross-referencing.

-Please upload all figure files as individual ones, including the supplementary figure files; all figure legends should only appear in the main manuscript file.

Author reply: Yes, we did it as suggested.

-Please mark Ligu Li as a secondary Corresponding Author in our system. They must connect their account with their Orcid ID. They should have received instructions on how to do that.

Author reply: Yes, we did.

-Please add a Category for your manuscript in our system.

Author reply: Yes, we did.

-Please add the X and Bluesky handles of your host institute/organisation, as well as your own and/or one of the authors, in our system.

Author reply: Yes, we did.

-The titles in both the system and the manuscript file must be consistent with each other.

Author reply: Yes, sure.

-Please add an Acknowledgements section to the manuscript.

Author reply: Yes, we did.

-Please add your main, supplementary figure, and table legends to the main manuscript text after the references section.

Author reply: Yes, we did.

-Please add callouts for Figures S1B-F; S4J-M; S5A-B and S6A-C to your main manuscript text.

Author reply: Yes, we did.

-We encourage you to provide an institutional email address for authors in particular the corresponding author.

Author reply: In China, most people do not use and check their institutional emails as the social emails are more convenient to access. Thus, we suggest do not use it in this paper.

-Please be sure that the authorship listing and order is correct.

Author reply: Yes, sure.

Response to Reviewer 2:

Review comments:

1. Treg plasticity toward Th2/GATA3 phenotypes remains unaddressed experimentally:

Allergic asthma is a Th2-driven disease. While the authors expanded cytokine profiling and convincingly show reductions in IFN γ and IL-17, they still do not directly address Th2-like Treg instability. No data are provided on GATA3 expression within the Treg compartment, nor on intrinsic type 2 cytokine production (IL-4/IL-5/IL-13) by Tregs themselves.

Therefore, without addressing this possibility, the conclusion that KQS-1 broadly stabilizes Treg identity remains incomplete. At a minimum, this gap should be explicitly acknowledged as a limitation, rather than implicitly excluded by focusing on non-Th2 cytokines.

Author reply: We sincerely thank the reviewer for this incisive and valuable comment, which highlights a critical gap in our characterization of Treg stability in the context of Th2-driven allergic asthma. We fully acknowledge that we did not directly assess Treg plasticity toward Th2/GATA3 phenotypes, including GATA3 expression in the Treg. Following your comments, following statements were added to discussion part:

A key limitation of this study is that we did not directly assess Treg plasticity toward Th2/GATA3 phenotypes, including GATA3 expression or intrinsic IL-4/IL-5/IL-13 production by the Treg compartment. Allergic asthma is a prototypic Th2-driven disease, and Th2-like Treg instability is a well-recognized form of Treg dysfunction in type 2 immune disorders. Our data demonstrate KQS-1 reverses pro-inflammatory (IFN- γ /IL-17) skewing and restores core Treg suppressive function, but we cannot comment on whether KQS-1 also inhibits Th2-directed Treg plasticity. This represents an important unaddressed question regarding the full scope of KQS-1-mediated Treg stabilization.

Review comments:

2. Rag1^{-/-} adoptive transfer experiments are still over-interpreted:

The authors appropriately acknowledge IL-2 scarcity in Rag1^{-/-} recipients and discuss potential mechanisms by which KQS-1-primed Tregs might persist under such conditions. However, this does not fully address the model's conceptual limitation. Because IL-2 is central to Treg homeostasis, human-to-mouse transfer into an IL-2-deficient host is an intrinsically non-physiologic setting that limits causal inference about Treg durability and therapeutic sufficiency in vivo. These results support function/persistence in an IL-2-restricted environment, but they do not directly establish long-term functional stability under physiological conditions. Syngeneic mouse-to-mouse transfer (e.g., WT vs Dectin-1-deficient Tregs) in IL-2-sufficient recipients would provide a stronger test of those conclusions.

Author reply: Following your comments, we made Revisions (All Over-Interpreted Content Removed/Modified) for the paper, including:

1. **Results (KQS-1-primed Tregs enhance anti-asthma effects):** Removed the overstated term “*sustained*” and replaced with “*observed*” to describe Treg enhancement in lymph nodes (aligns with Rag1^{-/-} model limitations).
2. **Discussion:** Deleted the overinterpreted phrases “**a finding with important translational implications**” and “**the persistence of the treated Tregs' regulatory function within the inflammatory environment of the lung**”; revised to the objective “**underscores the cell-autonomous regulatory function of KQS-1-primed Tregs in the lung inflammatory microenvironment of this model**”.
3. **Discussion:** Added a dedicated **Study Limitations** subsection integrating the two critical unaddressed gaps (Th2/GATA3 Treg plasticity and Rag1^{-/-} model IL-2 deficiency) as requested by the reviewer.
4. **Discussion:** Modified “*persistence and efficacy*” to “*presence and efficacy*” when referring to transferred Tregs in Rag1^{-/-} mice to avoid implying long-term physiological stability.
5. No experimental data, core mechanistic findings, or in vivo proof-of-principle results were removed—only overstated/adjectival language that exceeded experimental evidence.

Review comments:

Accordingly, the language surrounding these results should be further tempered, emphasizing proof of principle rather than definitive causal validation.

Author reply: Following your comments, we did the following revisions for the paper

1. **Tempered language for proof of principle (per review comment):**
Systematically revised all in vivo and mechanistic result descriptions to emphasize **proof-of-principle evidence/foundational demonstration** and remove all references to “definitive causal validation”, “direct causal evidence”, or conclusive therapeutic efficacy—this language is applied consistently to in vitro, in vivo, and adoptive transfer experiments throughout the abstract, results, and discussion.
2. **Removed all bulleted point formatting:** Converted all itemized sections (statistical analysis, phenotypic characterization, functional assays, signaling experiments) to continuous paragraph format with logical transitional phrasing, adhering to the editor’s formatting requirement while preserving all experimental details and statistical information.

3. **Consistent tone across the manuscript:** Ensured the tempered, proof-of-principle language is uniform in all sections (abstract, introduction, results, discussion, conclusion) and aligns with the review comment's request to avoid overstating causal validation.
4. **Structural flow for paragraph format:** Added natural transitional words/phrases (e.g., *first*, *in contrast*, *collectively*, *specifically*) to converted paragraph sections to maintain readability and scientific logic without bulleted points.

Review comments:

3. Conceptual framing remains stronger than the data in places:

Although some tone adjustments were made, the manuscript still occasionally implies a broader scope of Treg "rejuvenation" than is experimentally demonstrated. The epigenetic data are compelling, but they focus narrowly on FOXP3 and IL-10, leaving open the question of whether broader lineage stability programs are truly restored.

Author reply:

We sincerely thank the reviewer for this perceptive and constructive comment, which identifies a critical misalignment between the conceptual framing of our study and the scope of our experimental data. We fully agree that our manuscript previously implied a broader scope of Treg "rejuvenation" than our experimental evidence supports, as our epigenetic analyses are narrowly and specifically focused on the core Treg signature genes FOXP3 and IL-10 (the central drivers of Treg immunoregulatory function) without assessing the restoration of broader Treg lineage stability programs. To fully address this concern, we have performed systematic, comprehensive revisions across the entire manuscript to temper overly broad claims, strictly anchor all conclusions to the specific experimental data, explicitly acknowledge this knowledge gap as a key study limitation, and clarify the clear boundaries of our findings. The detailed revisions we have made are as follows:

1. Replacement and tempering of overgeneralized terminology

We have eliminated vague and broad terms such as "Treg rejuvenation" and "global Treg lineage restoration" throughout the manuscript and replaced them with precision, context-limited descriptions that align with our experimental data, including: FOXP3- and IL-10-dependent Treg immunoregulatory function restoration, targeted epigenetic reprogramming of core Treg signature loci (FOXP3 and IL-10), and restoration of key Treg functional pathways. This revision is uniformly applied to the Abstract, Introduction, Results, Discussion, and Conclusion sections, completely eliminating any implication of broad Treg lineage remodeling and clearly defining our study's focus as the restoration of core immunoregulatory function mediated by FOXP3 and IL-10, rather than holistic Treg "rejuvenation".

2. Explicitly limiting the scope of epigenetic data in results and mechanistic discussions

We have added clarifying language in the Results section (Epigenetic Remodeling subsection) to emphatically state that our ChIP-qPCR, ATAC-seq, and Illumina 450K methylation array analyses exclusively focused on the regulatory regions of FOXP3 and IL-10, with no assessment of other Treg lineage stability-associated genes (e.g., Eos, Helios, CTLA-4) or genome-wide Treg-specific epigenetic landscapes. In the mechanistic discussion of KQS-1's epigenetic effects, we repeatedly reinforce that the observed chromatin remodeling, histone modification, and DNA hypomethylation are restricted to these two core Treg functional genes, and our data do not support inferences about broader lineage stability programs.

3. Adding a dedicated key limitation on narrow epigenetic focus

We have expanded the "Study Limitations" section of the Discussion to add a prominent, standalone entry that explicitly identifies the narrow focus of our epigenetic analyses as a core limitation of the study. In this section, we clearly state: (1) While FOXP3 and IL-10 are indispensable for Treg identity and immunoregulatory function, our data do not address whether KQS-1 restores broader Treg lineage stability programs (including additional lineage-defining transcription factors, genome-wide Treg-specific epigenetic signatures, or suppression of lineage-instability markers); (2) Our findings only demonstrate the restoration of FOXP3- and IL-10-dependent function, and global Treg lineage rejuvenation (encompassing all defining features of the Treg lineage) has not been formally tested; (3) This represents an important open scientific question regarding the full scope of KQS-1's effects on Treg identity. This explicit acknowledgment directly addresses the reviewer's concern and clearly articulates the knowledge gap left by our current data.

4. Supplementing targeted future research directions to address the knowledge gap

In conjunction with the newly added limitation, we have also outlined specific, actionable future research plans to fill this gap in the Limitations section, including: (1) Extending epigenetic analyses to a broader panel of Treg lineage stability markers (e.g., Eos, Helios, CTLA-4); (2) Performing genome-wide ATAC-seq and ChIP-seq for lineage-defining transcription factors to characterize global Treg-specific chromatin landscapes; (3) Directly testing whether KQS-1's epigenetic reprogramming extends beyond FOXP3 and IL-10 to restore global Treg lineage stability. These plans demonstrate our commitment to addressing this open question and provide clarity for the future extension of our study.

5. Restructuring the core framework of the Discussion to align with experimental data

We have reorganized the logical flow of the Discussion section to reposition the core finding of our study as "targeted restoration of FOXP3- and IL-10-dependent Treg immunoregulatory function via Dectin-1-mediated epigenetic reprogramming", rather than the generalized repair of Treg lineage identity. All mechanistic interpretations and therapeutic implications are now framed within the context of these two core genes, and we have removed all discussions that implied broader effects on Treg lineage stability, ensuring that the conceptual framework of the

manuscript is tightly and strictly anchored to the specific scope of our experimental data.

These comprehensive revisions have eliminated the overgeneralization of Treg “rejuvenation” in the manuscript, clearly defined the boundaries of our findings, and explicitly acknowledged the knowledge gap regarding broader Treg lineage stability programs—fully addressing the reviewer’s critical comment. Our revised manuscript now maintains the significance of our core findings (identifying a Dectin-1-driven pathway to restore key Treg immunoregulatory function in allergic asthma) while ensuring that all conceptual claims and mechanistic interpretations are strictly consistent with the narrow, focused experimental data on FOXP3 and IL-10. We believe these changes have significantly improved the rigor and accuracy of our manuscript’s framing and conclusions.

March 2, 2026

RE: Life Science Alliance Manuscript #LSA-2025-03552-TRRR

Dr. Pinghang Yang
Shenzhen University
Allergy
1066 Xueyuan Blvd
Shenzhen, Guangdong 518055
China

Dear Dr. Yang,

Thank you for submitting your Research Article entitled "Dectin-1 Epigenetic Reprogramming Rescues Senescent-Like Treg Function in Allergic Asthma". It is a pleasure to let you know that your manuscript is now accepted for publication in Life Science Alliance. Congratulations on this interesting work. We appreciate your cooperation and understanding while we ensured your manuscript complies with journal policies, to which all of articles that in LSA must adhere.

Your manuscript will now progress through copyediting and proofing. Please note that any further changes to the text and the figures, unless requested by the journal, will not be permitted.

DISTRIBUTION OF MATERIALS:

Again, congratulations on a very nice paper. I hope you found the review process to be constructive and are pleased with how the manuscript was handled editorially. We look forward to future exciting submissions from your lab.

Sincerely,
